# CRITIQUE-CODER: ENHANCING CODER MODELS BY CRITIQUE REINFORCEMENT LEARNING

**Chi Ruan**[1]    **Dongfu Jiang**[1,2]    **Yubo Wang**[1,2]    **Wenhu Chen**[1,2]
[1]University of Waterloo    [2]Vector Institute
cruan059@uottawa.ca    wenhuchen@uwaterloo.ca

https://tiger-ai-lab.github.io/Critique-Coder

## ABSTRACT

Reinforcement Learning (RL) has emerged as a popular training paradigm, particularly when paired with reasoning models. While effective, it primarily focuses on generating responses and lacks mechanisms to explicitly foster critique or reflection. Several recent studies, like Critique-Fine-Tuning (CFT) and Critique-Guided-Distillation (CGD) have shown the benefits of explicitly teaching LLMs how to critique. Motivated by them, we propose Critique Reinforcement Learning (CRL), where the model is tasked with generating a critique for a given (question, solution) pair. The reward is determined solely by whether the final judgment label $c \in \{\texttt{True}, \texttt{False}\}$ of the generated critique aligns with the ground-truth judgment $c^*$. Building on this point, we introduce CRITIQUE-CODER, which is trained on a hybrid of RL and CRL by substituting 20% of the standard RL data with CRL data. We fine-tune multiple models (CRITIQUE-CODER) and evaluate them on different benchmarks to show their advantages over RL-only models. We show that CRITIQUE-CODER consistently outperforms RL-only baselines on all the evaluated benchmarks. Notably, our CRITIQUE-CODER-8B can reach over 60% on LiveCodeBench (v5), outperforming other reasoning models like DeepCoder-14B and GPT-o1. Beyond code generation, CRITIQUE-CODER also demonstrates enhanced general reasoning abilities, as evidenced by its better performance on logic reasoning tasks from the BBEH dataset. This indicates that the application of CRL on coding datasets enhances general reasoning and critique abilities, which are transferable across a broad range of tasks. Hence, we believe that CRL works as a great complement to standard RL for LLM reasoning.

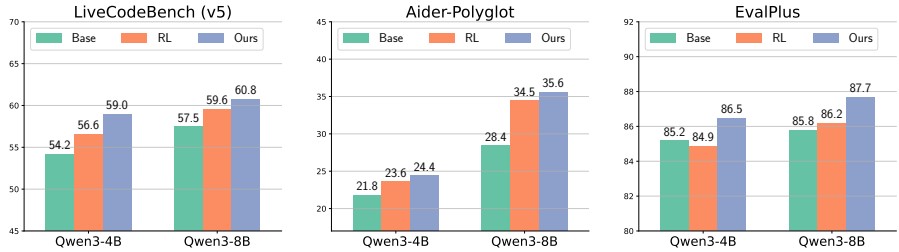

Figure 1: The effectiveness of our CRITIQUE-CODER, trained with a combination of CRL and RL data, compared to baselines and models trained solely on RL data, with both training and evaluation conducted under the think mode setting. EvalPlus denotes the average of 4 benchmarks: HumanEval, MBPP, and their corresponding plus version.

## 1    INTRODUCTION

Recent breakthroughs in complex reasoning across code generation, mathematical problem-solving, and logical deduction have been driven by large language models such as OpenAI's o1-o4 (Jaech et al., 2024a), DeepSeek-R1 (Guo et al., 2025), and Kimi-K1.5 (Team et al., 2025). A key factor behind these advances is the combination of reinforcement learning (RL) with chain-of-thought

(CoT) (Wei et al., 2022), which enables models to iteratively refine intermediate reasoning steps. Building on this foundation, research has increasingly focused on scaling reasoning abilities in code generation. For example, AceCoder (Zeng et al., 2025), HardTests (He et al., 2025), and Kod-Coder (Xu et al., 2025) developed automated large-scale code data generation pipelines and applied RL using reward models and test cases pass rewards, achieving notable performance gains. SWE-RL (Wei et al., 2025) pioneered the scaling of RL-driven LLM reasoning for real-world software engineering, leveraging an efficient rule-based reward system.

Standard reinforcement learning with verifiable reward (RLVR) has shown strong capabilities in improving models' problem-solving abilities. **However, this paradigm can hardly elicit the internal critique or reflection behavior on existing solutions**. Recently, there has been a line of work (Wang et al., 2025c; Gou et al., 2024; Xi et al., 2024; Kapusuzoglu et al., 2025; Wang et al., 2025a; Tang et al., 2024) aiming to explicitly teach LLMs to critique to unleash their reasoning ability. Inspired by this, we propose a new learning paradigm, **C**ritique **R**einforcement **L**earning (CRL), which incorporates this critique mechanism into RL and explicitly rewards models for accurate reflection. CRL not only optimizes problem-solving skills but also explicitly incentivizes its critique abilities via rewards for whether it can correctly judge a response's correctness. Specially, the model is prompted with a question–solution pair $[q; s]$ to predict a binarized judgment label $c \in \{\texttt{True}, \texttt{False}\}$, which is compared with the annotated label $c^*$ to derive a binary (verifiable) reward. This is in contrast to the standard RLVR algorithm, which incentivizes the model to predict a correct solution $s$ to a given query $q$, as shown in Figure 2. Different from CFT (Wang et al., 2025c), CRL incentivizes the model based on its self-generated judgment $c$ instead of using teacher-provided critique traces.

Based on the CRL paradigm, we develop CRITIQUE-CODER, a model trained to generate both high-quality coding solutions and critiques on existing solutions. We conducted a series of experiments with the GRPO algorithm (Shao et al., 2024). Specifically, we train QWEN3-4B and QWEN3-8B (Yang et al., 2025) on the filtered rStar seed dataset (Liu et al., 2025) using a hybrid framework that unifies CRL and standard RLVR. This hybrid approach enables the model to integrate the strengths of both paradigms: CRL helps the model develop critical thinking and reasoning abilities, while RLVR focuses on enhancing its problem-solving performance. We further adopted an iterative context lengthening approach Luo et al. (2025), where the training context length is extended from 16k to 32k tokens to better leverage the potential of long reasoning chains. By iteratively increasing the context length, the model first develops reasoning skills on shorter contexts, which can then be applied to longer ones.

CRITIQUE-CODER present consistent improvements across multiple benchmarks, illustrated in Figure 1. From QWEN3-4B, our CRITIQUE-CODER achieves 59.0 accuracy on LiveCodeBench (v5) (Jain et al., 2024), yielding +4.8 points over the base model and +2.4 points over the RL-only variant. Remarkably, it even surpasses QWEN3-8B by +1.5 points. On QWEN3-8B, CRITIQUE-CODER reaches 35.6 points on Aider-Polyglot, +7.2 points higher than baseline. It also reaches 60.8 points on LiveCodeBench (v5), which outperforms other reasoning models like DeepCoder-14B (Luo et al., 2025) and GPT-o1 (Jaech et al., 2024b). This showcases the effectiveness of CRL training. Furthermore, results on the logical reasoning benchmark BIG-Bench Extra Hard (Kazemi et al., 2025) demonstrate that CRITIQUE-CODER achieves strong transferable reasoning ability, surpassing both baseline and RL-trained models and yielding a +6.1 improvement over the base model. We also find that CRL is more effectively utilized as a complement to RL rather than serving as an alternative. This is because CRL training primarily focuses on critiquing question–solution pairs without generating actual solutions. Our ablation study in Table 5 confirms a 20% mix ratio as a best practice.

In summary, we introduce CRL, a novel reinforcement learning (RL) training framework that incorporates critique learning within the RL paradigm. This novel learning approach enhances the model's critique and reasoning abilities, addressing the lack of critique and reflection incentives typically found in standard RL frameworks. Building on this foundation, we introduce CRITIQUE-CODER, a model combining CRL and RL to leverage the strengths of both. CRL fosters critical thinking and reasoning, while RL focuses on optimizing problem-solving. Compared to baseline models and those trained exclusively with RL, CRITIQUE-CODER shows superior performance across coding datasets of varying difficulty. Furthermore, the model demonstrates transferable general reasoning abilities, as evidenced by its strong performance on logic reasoning benchmarks.

## 2 METHOD

This section introduces the training process of CRITIQUE-CODER, which combines standard reinforcement learning and critique reinforcement learning for code generation. We first present the problem formulation and optimization method, followed by dataset construction and the overall training procedure.

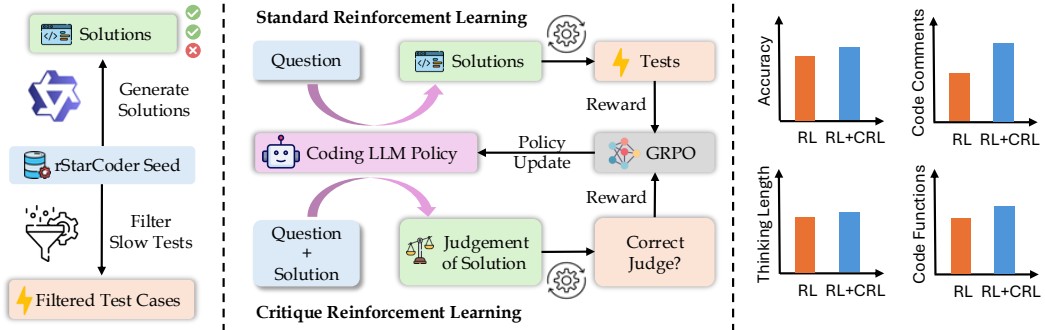

Figure 2: Comparison between CRL and Standard RL. Standard RL generates solutions based on input questions and evaluates them by executing test cases, while CRL critiques the solution for the paired question and compares the resulting conclusion with the GT to determine its correctness. Experiment shows that RL+CRL can improve not only accuracy, but also the code quality.

### 2.1 PRELIMINARY

**Problem Definition.** CRITIQUE-CODER incorporates two complementary training frameworks for LLMs. The first follows the standard RL setting: given a question $q$, the policy $\pi_\theta$ samples $n$ candidate solutions $\{s_i\}_{i=1}^n$; each $s_i$ is evaluated on the annotated test cases $\mathcal{T}$ to compute its pass rate, which serves as the reward signal $R_{\text{rl},i}$.

To complement this solution-level feedback, we introduce Critique Reinforcement Learning (CRL), which provides binary correction signals on question–solution pairs. Specially, given an annotated dataset $\mathcal{D} = \{([q_k; s_k], c_k^*)\}_{k=1}^N$, where each pair ($[q; s]$) consists of a question and a solution with an associated binary judgment label $c^* \in \{0, 1\}$, the policy $\pi_\theta$ is trained to generate $n$ predictions $\{c_i\}_{i=1}^n$ indicating whether $s$ satisfies the requirement posed by $q$. The reward $R_{\text{crl},i}$ is derived from the comparison between $c_i$ and $c^*$.

Finally, two reward signals $R_{\text{rl},i}$ from RL and $R_{\text{crl},i}$ from CRL are combined together to update policy parameters $\theta$ using GRPO. This unified optimization enables the model to benefit from both critique-guided learning and task-oriented learning, fostering more critical and reflective learning.

**Group Relative Policy Optimization (GRPO).** We now detail GRPO (Shao et al., 2024), the optimization algorithm used to update model parameters. In contrast to PPO (Schulman et al., 2017), GRPO enhances performance by leveraging relative performance-based updates, which yield more stable and efficient policy refinement. The formal definition of GRPO is provided below:

$$\mathcal{J}(\theta) = \mathbb{E}\left[\frac{1}{G}\sum_{i=1}^{G}\frac{1}{|o_i|}\sum_{t=1}^{|o_i|}\min\left(\rho_{i,t}\hat{A}_{i,t},\, \text{clip}(\rho_{i,t}, 1-\epsilon, 1+\epsilon)\hat{A}_{i,t}\right) - \beta\,\mathbb{D}_{\text{KL}}\left(\pi_\theta\|\pi_{\text{ref}}\right)\right],$$

$$\text{where } \rho_{i,t} = \frac{\pi_\theta(o_{i,t} \mid x, o_{i,<t})}{\pi_{\theta_{\text{old}}}(o_{i,t} \mid x, o_{i,<t})}. \tag{1}$$

In the above equation, $\rho_{i,t}$ denotes the probability ratio of generating output $o_{i,t}$ under the new policy $\pi_\theta$ and old policy $\pi_{\theta_{\text{old}}}$, $\hat{A}_{i,t}$ represents the calculated advantage within each output group, and $\mathbb{D}_{\text{KL}}$ regularizes the optimization by measuring the divergence between $\pi_\theta$ and the reference policy $\pi_{\theta_{\text{ref}}}$, which can prevent the policy from drifting too far away.

Table 1: Dataset statistics of rStar-Coder seed dataset before and after test-case filtering to save the verification time.

| Metric | Before | After |
|---|---|---|
| Num of Questions | 29,365 | 23,069 |
| Avg Test Cases | 87 | 24 |
| Median Test Cases | 48 | 30 |
| Avg Test Case Input Chars | 96,208 | 40 |

Table 2: Dataset difficulty statistics using QWEN3-4B (Thinking) with temperature=0.6, top-p=0.95, and top-k=20

| Metric | Value |
|---|---|
| Pass@1 | 43.72% |
| Pass@2 | 49.05% |
| Pass@4 | 52.98% |
| Avg Tokens / Solution | 13,732 |

In our training scenario, the policy input $x$ can be either a single question $q$ from RL or a question–solution pair ($[q; s]$) from CRL. These two input modalities give rise to distinct reward signals, solution-level rewards $R_{\text{rl},i}$ and critique-level rewards $R_{\text{crl},i}$. Both signals are aggregated in the computation of the advantage $\hat{A}_{i,t}$, which makes the GRPO update in our framework fundamentally different from standard RL: the advantage estimation is jointly shaped by task outcomes and critique guidance, allowing the policy to align with both execution correctness and reflective judgment.

## 2.2 DATASET CONSTRUCTION

To evaluate the efficacy of CRL, the first step is to build a reliable CRL dataset. The construction process is detailed in the following steps.

**RL Dataset Selection.** We construct our CRL dataset from the human-seeded RL dataset of rStar-Coder (Liu et al., 2025), which contains a large number of test cases collected from both human-written and verified generated data. To generate these test cases, RStar-Coder employs utility functions to generate test case inputs across a wide range of scales, reaching up to $10^5$ for challenging cases. As a result, many questions in the original dataset include an excessive number of test cases (often exceeding 100 per problem), and some individual cases are extremely long (over 10,000 tokens). Such characteristics substantially increase verification time during RL training. To improve efficiency and consistency, we filter the data by discarding test cases longer than 200 tokens and randomly sampling 30 cases for each problem. Table 1 reports dataset statistics before and after filtering, showing a significant reduction in both test case length and volume. Specifically, the average input characters decrease from 96,208 to 40, and the average number of cases drops from 87 to 24. This reduction greatly shortens test case evaluation time during training, resulting in a more efficient learning process. To assess dataset difficulty, we evaluate QWEN3-4B (Yang et al., 2025) on the filtered dataset, shown in Table 2. The model achieves 43.72% at Pass@1 and 52.98% at Pass@4, indicating a moderate difficulty level—solvable in part, yet leaving significant headroom for further progress. This makes the dataset well-suited for RL training under the GRPO algorithm, where advantage is computed within groups.

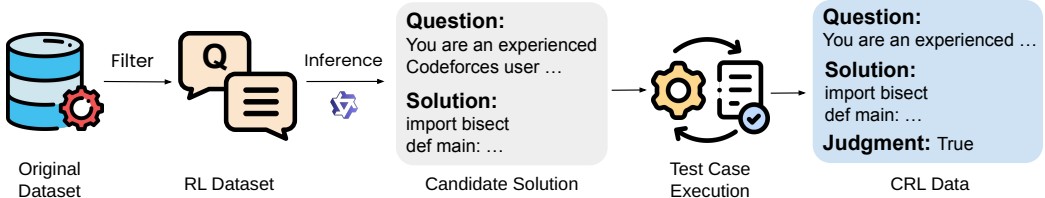

Figure 3: Critique data generation. This process involves generating candidate solutions and annotating their judgment in the CRL dataset based on the pass rate over test cases.

**Critique Data Generation.** Figure 3 illustrates the critique data generation workflow. For each problem, we prompt QWEN3-CODER-30B-A3B-INSTRUCT (Yang et al., 2025) to generate outputs, from which we extract code blocks as candidate solutions. Despite its larger parameter scale, QWEN3-CODER-30B-A3B-INSTRUCT lacks explicit reasoning capabilities, and its effec-

tive problem-solving performance is not stronger than that of smaller reasoning-enabled models such as QWEN3-4B (Thinking). Empirically, QWEN3-CODER-30B-A3B-INSTRUCT achieves a pass@1 of 32.20% on our dataset, compared to 43.72% for QWEN3-4B (Thinking). Owing to this relatively weak performance, the critique data generation process does not constitute a distillation setup, as the data are not produced by a stronger teacher model. Empty code blocks are discarded to ensure that only valid programs are retained for evaluation. Each candidate solution is then executed on the test cases from the filtered dataset, and its pass rate is computed to determine its judgment. A practical challenge is that certain test cases exhibit excessively long execution times, which may cause timeouts and lead to the erroneous classification of correct solutions as failures. To relax this constraint, we adopt a pass rate threshold of 80%: candidate solutions are labeled as `True` if their test pass rate exceeds this percentage, and as `False` otherwise.

**Hybrid Data Integration.** In our case, training exclusively on critique-oriented data biases the model toward evaluative behaviors, encouraging it to focus on judging or analyzing candidate solutions rather than directly producing task-oriented outputs, which in turn degrades its ability to generate complete answers in evaluation tasks. Standard RL data, by contrast, explicitly reinforces end-to-end solution generation through task-level rewards but does not expose the model to reflective or critique-based reasoning patterns. To mitigate this imbalance, we construct a hybrid dataset that combines both CRL and standard RL data, allowing the model to jointly learn evaluative judgment and direct solution synthesis within a unified training process. Concretely, we randomly assign 20% of the data from the dataset to be CRL data, with the remaining 80% consisting of standard RL data. Such a configuration not only mitigates the risk of format shift caused by overexposure to critique-style supervision but also improves the robustness of the learning process by exploiting the complementary strengths of CRL and RL training paradigms.

## 2.3 TRAINING

**Training procedure.** Algorithm 1 presents the training procedure of CRITIQUE-CODER, which integrates CRL and standard RL within a unified framework. The policy model is initialized from a pretrained checkpoint $\pi_{\theta_{\text{init}}}$ and trained on a hybrid dataset containing both CRL samples with judgment labels and RL samples with associated test cases. At each training step, the policy samples $G$ candidate outputs for each instance under either the CRL or RL setting. For CRL data, the model produces multiple critique predictions, parses the judgment from the `\conclusion{}` field, and computes rewards by comparing the predicted judgments with the ground-truth labels. For RL data, the model generates multiple solution candidates, extracts the code blocks enclosed by ` ```[code]``` `, and evaluates them against the provided test cases to obtain execution-based rewards. These rewards are then transformed into advantage estimates and jointly used to update the policy parameters through GRPO.

---

**Algorithm 1** Training procedure of CRITIQUE-CODER

---

**Input** dataset $\mathcal{D} = \{([q_i; s_i], c_i^*)\}_{i=1}^{N_1} \cup \{(q_j, t_j)\}_{j=1}^{N_2}$, policy $\pi_\theta$
1: Initialize policy model $\pi_\theta \leftarrow \pi_{\theta_{\text{init}}}$
2: **for** each step **do**
3:      Sample a batch $\mathcal{B} \subset \mathcal{D}$
4:      **for** each data instance $d \in \mathcal{B}$ **do**
5:          **if** $d = ([q; s], c^*)$                       ▷ CRL data with judgment **then**
6:              Sample $G$ outputs $\{o_i\}_{i=1}^G \sim \pi_\theta([q; s])$
7:              Parse each $o_i$ to extract judgment $c_i$ inside `\conclusion{}`
8:              Compute reward $R_{\text{crl},i}(c_i, c^*)$ for each $c_i$
9:          **else if** $d = (q, t)$                    ▷ RL data with test cases **then**
10:             Sample $G$ outputs $\{o_i\}_{i=1}^G \sim \pi_\theta(q)$
11:             Parse each $o_i$ to extract solution $s_i$ enclosed by ` ```[code]``` `
12:             Evaluate $s_i$ on test cases $t$, obtain reward $R_{\text{rl},i}(s_i, t)$
13:          **end if**
14:      **end for**
15:      Compute $\hat{A}_{i,t}$ from reward $R_i$, where $R_i = R_{\text{crl},i}(c_i, c^*)$ for CRL or $R_{\text{rl},i}(s_i, t)$ for RL.
16:      Update the policy model $\pi_\theta$ with GRPO ( Equation 1)
17: **end for**

---

**Reward function.** As specified in Algorithm 1, rewards are computed from two data sources: CRL and RL. For CRL samples, the model is prompted to store the final judgment in \conclusion{}, from which the predicted label $c$ is extracted. A reward of 1 is assigned if $c$ matches the ground truth $c^*$; otherwise, including the case where the prediction is missing, the reward is 0. For RL samples, the reward is defined as the pass rate across test cases. Formally,

$$R_{\text{crl}}(c, c^*) = \begin{cases} 1, & \text{if } c = c^*, \\ 0, & \text{otherwise,} \end{cases}, \quad R_{\text{rl}}(s, \mathcal{T}) = \frac{K}{N} \qquad (2)$$

where $\mathcal{T}$ is the set of $N$ test cases and $K$ denotes the number of cases successfully solved by the model's output $s$. Thus $R_{\text{rl}} \in [0, 1]$, with larger values indicating more reliable solutions. At the batch level, each instance $i$ receives its reward according to its data type:

$$R_i = \begin{cases} R_{\text{crl}}(c_i, c_i^*), & \text{if } i \in \mathcal{B}_{\text{CRL}}, \\ R_{\text{rl}}(s_i, \mathcal{T}_i), & \text{if } i \in \mathcal{B}_{\text{RL}}. \end{cases} \qquad (3)$$

Here $\mathcal{B}_{\text{CRL}}$ and $\mathcal{B}_{\text{RL}}$ denote the CRL and RL subsets within the batch, respectively.

## 3 EXPERIMENT

This section presents a comprehensive empirical evaluation to demonstrate the effectiveness of CRITIQUE-CODER. We report training and evaluation settings, compare against baseline and RL-trained models on multiple coding benchmarks, and further analyze transferability, test-time scaling, and ablation results.

### 3.1 TRAINING SETUP

We conducted experiments on two models, QWEN3-4B and QWEN3-8B (Yang et al., 2025), in thinking mode. Following a two-phase training strategy similar to DeepCoder (Luo et al., 2025), we set the maximum response length to 16k in the first phase and increase it to 32k once the rewards have stabilized. During training, two rule-based rewards are employed, each tailored to a different data type. For CRL data, the reward is 1.0 if the prediction matches the ground truth (GT) and 0.0 otherwise; additionally, during the 16k phase, this reward is scaled by a factor of 0.8 to reduce its dominance relative to RL signals. For RL data, the reward corresponds to the pass rate over test cases, ranging from 0.0 to 1.0. For standard RL training, we adopt the thinking prompt used in the Qwen3 paper on LiveCodeBench, while the CRL training prompt is provided in Appendix A. Throughout training, we apply the GRPO algorithm (Shao et al., 2024), which provides improved stability and efficiency compared to PPO (Schulman et al., 2017). The hyperparameters are set as follows: a batch size of 128, a learning rate of 1e-6, and 8 sampled outputs per prompt. To encourage exploration while stabilizing entropy, the clipping ratio is asymmetric, with an upper bound of 0.3 and a lower bound of 0.2. We trained the model on the entire dataset for one epoch, and selected the best-performing checkpoint using the LiveCodeBench (v5) as the validation set.

### 3.2 EVALUATION SETUP

To evaluate and compare our training results, we utilized four different benchmarks: EvalPlus (Liu et al., 2023) (an average of HumanEval (Chen et al., 2021), HumanEval+, MBPP (Austin et al., 2021), and MBPP+), BigCodeBench-Instruct (Zhuo et al., 2024), Aider-Polyglot (Aider, 2024), and LiveCodeBench (v5, 2024.10–2025.02) (Jain et al., 2024). These benchmarks cover a diverse range of coding tasks, enabling a comprehensive assessment of the model's code generation ability.

For the sampling configuration, we follow the thinking mode settings reported in the original Qwen3 paper (Yang et al., 2025), with a temperature of 0.6, a top-p of 0.95, a top-k of 20, and a maximum output length of 32,768 tokens. The same configuration is applied consistently across all evaluation tasks. For LiveCodeBench, we adopt the official evaluation prompt used in Qwen3 thinking mode to ensure consistency. We further compare our model with several strong coding baselines, including DeepSeek-R1-distill-14B (Guo et al., 2025), DeepCoder (Luo et al., 2025), DeepSeek-V2.5 (DeepSeek-AI, 2024), and GPT-o1 (Jaech et al., 2024b) with high reasoning effort, and report results from their official publications or recommended evaluation settings.

Table 3: CRITIQUE-CODER performance compared with baseline and models trained with standard RL. The RL training was conducted on the filtered rStar-Coder seed dataset, and the CRL training was carried out by converting 20% of the data into CRL for fair comparison.

| Model | EvalPlus | BigCodeBench-I Full | Hard | Aider-Polyglot | LiveCodeBench v5 | AVG |
|---|---|---|---|---|---|---|
| AceCoder-7B | 82.7 | 43.3 | 19.6 | - | - | - |
| DeepSeek-R1-Distill-14B | 82.4 | 38.1 | 20.9 | 18.6 | 53.0 | 42.6 |
| DeepCoder-14B | 85.3 | 38.2 | 18.2 | 18.4 | 60.6 | 44.1 |
| DeepSeek-V2.5-238B | 83.8 | 48.9 | 27.0 | 17.8 | 42.6 | 44.0 |
| GPT-o1 | 88.6 | 50.4 | 28.4 | 61.7 | 59.5 | 57.7 |
| Baseline = Qwen3-4B (Thinking) | | | | | | |
| Baseline | 85.2 | 42.0 | 20.9 | 21.8 | 54.2 | 44.8 |
| Qwen3-4B-RL | 84.9 | 40.6 | **23.0** | 23.6 | 56.6 | 45.7 |
| CRITIQUE-CODER | **86.5** | **43.1** | **23.0** | **24.4** | **59.0** | **47.2** |
| Δ (Ours-Baseline) | +1.3 | +1.1 | +2.1 | +2.6 | +4.8 | +2.4 |
| Baseline = Qwen3-8B (Thinking) | | | | | | |
| Baseline | 85.8 | 44.6 | 23.6 | 28.4 | 57.5 | 48.0 |
| Qwen3-8B-RL | 86.2 | 44.5 | 24.3 | 34.5 | 59.6 | 49.8 |
| CRITIQUE-CODER | **87.7** | **46.6** | **27.0** | **35.6** | **60.8** | **51.5** |
| Δ (Ours-Baseline) | +1.9 | +2.0 | +3.4 | +7.2 | +3.3 | +3.5 |

## 3.3 MAIN RESULTS

**Gains over Base Models.** Compared with the base models, CRITIQUE-CODER leads to consistent and notable improvements across benchmarks of varying difficulty levels and evaluation protocols. On QWEN3-4B, for example, the LiveCodeBench score of CRITIQUE-CODER increases from 54.2 to 59.0, yielding a substantial gain of +4.8 and even surpassing the larger QWEN3-8B baseline by +1.5 points. On the Aider-Polyglot benchmark, which evaluates multi-language programming ability, CRITIQUE-CODER achieves a +7.2 improvement over the QWEN3-8B baseline, despite being trained solely on Python data using CRL. Importantly, these gains are consistently reflected across all evaluated benchmarks and model scales, and also translate into higher overall average scores (+2.4 for 4B and +3.5 for 8B), indicating that the improvements are broad-spectrum rather than limited to specific tasks or datasets.

**Advantages over RL-trained Models.** Under identical datasets and training configurations, replacing part of the RL data with CRL consistently yields superior results across all benchmarks. On QWEN3-4B, CRITIQUE-CODER exceeds the Qwen3-4B-RL by +2.4 points on LiveCodeBench and improves the overall benchmark average by +1.5 points. On QWEN3-8B, it outperforms the Qwen3-8B-RL counterpart by +2.7 points on BigCodeBench-Hard, contributing to an average gain of +1.7 points across all benchmarks. These findings highlight that CRL brings complementary benefits over RL, enabling CRITIQUE-CODER to achieve more robust and consistent improvements. We further analyze the test outputs of CRL and standard RL on LiveCodeBench, as illustrated in Figure 4. The results show that CRL generates longer reasoning traces in the think blocks, indicating more extensive deliberation and reflection, and confirming that CRL indeed enhances the model's reasoning and critique capabilities. It also incorporates markedly more explanatory comments within the generated code, indicating stronger tendencies toward self-explanation.

**Comparison with Frontier Models.** Despite their relatively small parameter sizes, our models demonstrate strong absolute performance when compared with frontier systems. CRITIQUE-CODER-4B significantly outperforms DeepCoder-14B (Luo et al., 2025) across all evaluated benchmarks while using only 28% of its parameters, highlighting the parameter efficiency of our approach. CRITIQUE-CODER-8B remains competitive with other strong models and only lags behind GPT-o1 (Jaech et al., 2024b) on Aider-Polyglot, which can be largely attributed to its lack of explicit optimization for non-Python languages. On EvalPlus, CRITIQUE-CODER-4B achieves an impressive score of 86.5, closely trailing GPT-o1, while CRITIQUE-CODER-8B further improves this result to 87.7, demonstrating that our method scales effectively with model size.

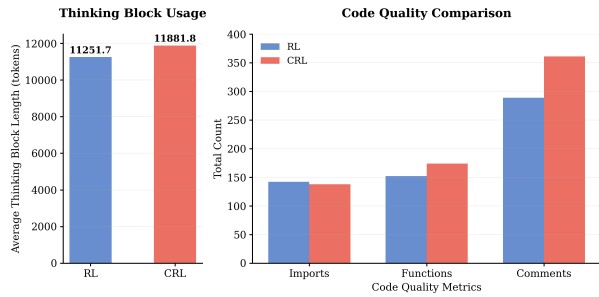
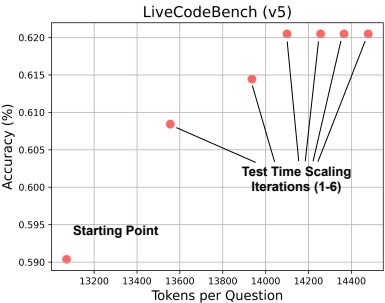

Figure 4: Analysis of the generations on the Live-CodeBench (v5) problems. Results show that CRL can elicit better reasoning behavior and coding quality.

Figure 5: Test-time scaling performance of CRITIQUE-CODER-4B on LiveCodeBench (v5)

Table 4: Performance comparison on four BIG-Bench Extra Hard (BBEH) logic reasoning subtasks

| Model | Time Arithmetic | DisambiguationQA | Zebra Puzzles | BoardgameQA | AVG |
|---|---|---|---|---|---|
| Baseline = Qwen3-4B (Thinking) | | | | | |
| Baseline | 40.5 | 43.3 | 36.5 | 66.5 | 46.7 |
| Qwen3-4B-RL | 45.0 | 43.3 | 40.5 | 66.5 | 48.8 |
| CRITIQUE-CODER | **48.5** | **47.5** | **44.0** | **71.0** | **52.8** |
| Δ (Ours-Baseline) | +8.0 | +4.2 | +7.5 | +4.5 | +6.1 |

## 3.4 TRANSFERABLE REASONING ABILITY

To examine whether the critique and reasoning abilities learned by CRITIQUE-CODER extend beyond coding tasks, we evaluate the model on the BIG-Bench Extra Hard (BBEH) logic reasoning benchmark (Kazemi et al., 2025). As shown in Table 4, CRITIQUE-CODER consistently outperforms both the baseline QWEN3-4B and its RL-trained variant across all four reasoning subtasks, achieving an average improvement of +6.1 points over the base model. The gains are especially pronounced on more structured and constraint-intensive tasks such as Time Arithmetic (+8.0) and Zebra Puzzles (+7.5), which require multi-step logical consistency rather than surface-level pattern matching. Moreover, despite being optimized with reinforcement learning, Qwen3-4B-RL shows only limited improvements, whereas CRITIQUE-CODER achieves an additional +4.0 average gain and outperforms it by +4.5 points on BoardgameQA. Notably, we use the same inference configuration for all models, suggesting that the gains are driven by critique-enhanced training rather than evaluation-time settings. Overall, these results indicate that critique-enhanced training provides additional benefits over standard RL optimization and effectively improves reasoning abilities that transfer beyond coding tasks.

## 3.5 TEST TIME SCALING

We further evaluate CRITIQUE-CODER-4B with sequence test-time scaling (budget forcing) (Muennighoff et al., 2025), which relaxes the constraint on reasoning tokens at inference to enable longer reasoning trajectories. As shown in Figure 5, increasing the test-time computation budget leads to a consistent and monotonic improvement on LiveCodeBench (v5): starting from the default setting without test-time scaling (approximately 13k tokens per question and 59.0 accuracy), the model steadily improves as the number of scaling iterations increases, reaching an accuracy of 62.0 after four iterations and achieving a +3.0 absolute gain over the no-scaling baseline. Further increases in the reasoning budget yield marginal but stable improvements, indicating diminishing returns. Overall, these results demonstrate that test-time scaling is an effective mechanism for enhancing the performance of a compact 4B code model without modifying its parameters. We note that this improvement is specific to sequence-based test-time scaling; in contrast, parallel test-time scaling does not yield gains, which we analyze in detail in subsection 3.7.

Table 5: Impact of different CRL data proportion. All datasets are derived from the filtered rStar-Coder seed dataset, with varying proportions of RL data converted into CRL data.

| Model | EvalPlus | BigCodeBench-I | | Aider-Polyglot | LiveCodeBench v5 | AVG |
| | | Full | Hard | | | |
|---|---|---|---|---|---|---|
| Baseline = Qwen3-4B (Thinking) | | | | | | |
| 0% of CRL Data | 84.9 | 40.6 | **23.0** | 23.6 | 56.6 | 45.7 |
| 50% of CRL Data | **86.5** | 42.4 | 22.3 | 24.0 | 56.0 | 46.2 |
| 100% of CRL Data | 85.2 | 41.6 | 17.6 | 21.3 | 56.6 | 44.5 |
| 20% of CRL Data (Ours) | **86.5** | **43.1** | **23.0** | **24.4** | **59.0** | **47.2** |

## 3.6 ABLATION STUDY

To study the impact of the CRL data ratio, we conduct an ablation study by replacing different proportions of RL data with CRL data while keeping the total amount of training data and optimization settings fixed. All training samples are derived from the same filtered rStar-Coder seed dataset, and CRL data are obtained by converting a subset of the original RL samples following the same procedure. As shown in Table 5, introducing a moderate amount of CRL data consistently improves performance over the pure RL baseline. Among all configurations, the model trained with 20% CRL data achieves the best overall results, increasing the average score from 45.7 to 47.2 (+1.5). This setting yields the strongest or tied-best performance on most benchmarks, including EvalPlus, BigCodeBench-I (Full), Aider-Polyglot, and LiveCodeBench v5, demonstrating that CRL can effectively complement RL training.

As the CRL ratio increases further, the performance gains gradually diminish. Although the 50% CRL setting remains competitive on average, it shows degradation on more challenging subsets such as BigCodeBench-I (Hard). When RL data are completely replaced by CRL data, performance drops significantly across multiple benchmarks, resulting in an average score (44.5) lower than the baseline. These results suggest that CRL should be used as a complementary signal rather than a replacement for RL data. We hypothesize that excessive reliance on CRL may bias the model toward judgment-oriented behaviors, introducing a mismatch with inference-time requirements that emphasize long-horizon solution generation, particularly on harder benchmarks. Overall, maintaining an appropriate balance between RL and CRL data is crucial, with 20% CRL providing the most favorable trade-off in our experiments.

## 3.7 LIMITATIONS IN SELF-CRITIQUE

Although incorporating CRL substantially improves the model's reasoning and critique-related capabilities during generation, it does not enable reliable post-hoc self-critique when only the final answer is available. To examine this limitation, we implemented critique-based parallel test-time scaling on CRITIQUE-CODER. Specifically, on LiveCodeBench v5, CRITIQUE-CODER-4B was prompted to generate 10 candidate solutions per problem. Each solution, together with the original question, was then fed back into the model for critique, with 64 critique samples generated per solution. Candidate solutions were scored by the number of `True` critiques they received, with ties broken by selecting the solution with the shortest critique thinking token length. This procedure did not lead to performance improvements.

This outcome is not specific to our method and is consistent with prior findings (Huang et al., 2023; Valmeekam et al., 2023) showing that large language models cannot reliably evaluate or refine their own final outputs without access to intermediate reasoning trajectories or additional external signals. Importantly, this result does not contradict the effectiveness of CRL. The performance gains from CRL do not arise from post-hoc self-critique on completed answers; instead, CRL improves performance by internalizing critique-based judgment during training, enabling more thorough reasoning and refinement *before* the final answer is produced. Consequently, once the reasoning process is removed and only the final output is provided, the model naturally fails to further critique or improve it, in line with established limitations in the literature.

# 4 RELATED WORKS

## 4.1 CRITIQUE LEARNING

The idea of leveraging critiques for improving model reasoning has been explored in several lines of research. One direction is self-correction (Bai et al., 2022; Gou et al., 2023; Madaan et al., 2023; Shinn et al., 2023), where models iteratively evaluate and revise their own outputs. Although such methods are promising, subsequent studies have questioned their robustness and consistency (Huang et al., 2023; Valmeekam et al., 2023). Another line involves reward models (Uesato et al., 2022; Wang et al., 2023; Lightman et al., 2023), which act as learned evaluators that assign quality scores to either final outputs or intermediate reasoning steps, thereby guiding reinforcement learning to enhance reasoning capabilities. More recently, Critique Fine-Tuning (CFT) (Wang et al., 2025c;b) explicitly trains models to critique candidate solutions, demonstrating improved reasoning ability.

Our approach is most related to CFT. Unlike CFT, which directly optimizes the model to imitate the critique reasoning process, CRL instead encourages the model to actively explore and learn from the correctness of its final judgments, thereby combining the benefits of critique reasoning with reinforcement feedback.

## 4.2 REINFORCEMENT LEARNING FOR CODE GENERATION

Code generation, a core capability of LLMs, has received considerable attention. CodeRL (Le et al., 2022) introduces a pioneering RL framework for code generation, employing an actor–critic architecture to encourage functionally correct outputs. Building on this foundation, PPOCoder (Shojaee et al., 2023) incorporates the PPO algorithm to further stabilize and improve training. Moreover, RLEF (Gehring et al., 2024) advances the paradigm by explicitly leveraging execution feedback during synthesis. More recently, AceCoder (Zeng et al., 2025) proposes a scalable pipeline that automatically constructs question–test case pairs from code data to facilitate RL training.

## 4.3 CHAIN-OF-THOUGHT REASONING.

Recent advances in reasoning language models (RLLMs) show that extended chain-of-thought (CoT) reasoning substantially improves performance on tasks like coding and mathematics. OpenAI o1 (Jaech et al., 2024a) and DeepSeek R1 (Guo et al., 2025) exemplify this trend by using inference-time scaling, where models iteratively explore and reflect before converging on a solution. Building on this, KIMI K1.5 (Team et al., 2025) simplifies the reinforcement learning framework while incorporating long-context scaling and enhanced policy optimization, further advancing reasoning efficiency. More recently, Qwen3 (Yang et al., 2025) combines a *thinking mode* for reasoning with a *non-thinking mode* for fast responses, switching between them to balance latency and performance.

# 5 CONCLUSION

In this paper, we introduce Critique Reinforcement Learning (CRL), a novel reinforcement learning paradigm that integrates critique learning into the RL by incorporating feedback on the correctness of critiques predicted by the model. Building on this foundation, we developed CRITIQUE-CODER, trained on a mix of RL and CRL data. On LiveCodeBench v5, CRITIQUE-CODER-4B achieves a score of 59.0, outperforming the baseline by +4.8 points and the RL-only model by +2.4 points. In addition to coding tasks, CRL also enhances general reasoning ability. On the BBEH logical reasoning benchmark, CRITIQUE-CODER shows substantial improvements, surpassing the baseline and RL-trained models by +6.1 and +4.0 points on average across four subtasks. These results demonstrate that CRL not only boosts critique and reflection abilities over standard RL but also enables these capabilities to extend beyond coding domains. However, ablation studies reveal that training exclusively on CRL data yields poorer performance than RL alone, since CRL focuses on generating critiques rather than task-oriented solutions, leading to a mismatch with evaluation requirements. Therefore, rather than substituting RL, CRL serves as a powerful complement to it. Taken together, our findings demonstrate that CRL enhances standard RL by endowing models with stronger critique and reasoning abilities—capabilities that manifest not only in coding tasks but also transfer effectively to broader reasoning domains.

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

APPENDIX

## A CRL TRAINING PROMPT

We provide the prompt template used for constructing CRL training data.

Table 6: Prompt for constructing CRL training data

---

You will be given a question (problem specification) and a submitted solution. Your task is to determine whether the solution is correct and fully satisfies the specification.

Question: {question}

Solution: {solution}

Conclude with \conclusion{T} for correct, \conclusion{F} for wrong.

---

