# OpenReview forum: "Critique-Coder: Enhancing Coder Models by Critique Reinforcement Learning"
_ICLR.cc/2026/Conference — ICLR 2026 Poster_

### Official Review · Reviewer_qbh7 · 2025-10-25

**Soundness:** 4
**Presentation:** 4
**Contribution:** 3
**Rating:** 8
**Confidence:** 4

**Summary:**

This paper proposes integrating *critique data* into reinforcement learning from verifiable rewards (RLVR) for large language models (LLMs). The critique rewards are binary (1–0), indicating whether the model’s judgment is correct relative to a reference solution. The study focuses on coding tasks, where solutions can be automatically verified using unit tests. Experimental results demonstrate that incorporating critique data leads to significant performance improvements on coding benchmarks such as **LiveCodeBench**.

**Strengths:**

* The improvements over baseline models are substantial.
* The paper is clearly written and easy to follow.
* The experiments are carefully designed, with appropriate baselines and ablation studies.
* Overall, the proposed approach is simple yet effective, and the empirical results are convincing.

**Weaknesses:**

* The idea of using critique data is not entirely novel, as similar concepts have been explored in supervised fine-tuning (SFT). However, to the best of my knowledge, this is the first work to apply critique data within an RL setting with verifiable rewards.
* The contribution is somewhat incremental, as it mainly involves augmenting an existing RL framework with additional data.

**Questions:**

The authors note that

> Although incorporating CRL enhances the model’s reasoning and critique abilities, it does not exhibit self-critique capability.

They further present an experiment showing that having the model critique its own solutions does not improve performance. Typically, increasing compute at test time tends to yield better results. Why does this not hold in this case? Could the authors elaborate further on the intuition or mechanisms behind this observation?

---

> ### Author Response · Authors · 2025-11-14
>
> We sincerely thank the reviewer for the positive and encouraging feedback. We are glad that the clarity of presentation and the effectiveness of our approach were well recognized.
>
> Regarding the reviewer’s question about why self-critique does not improve performance despite increased test-time compute, in our experiments, the model only had access to the final code output during critique, without observing the intermediate reasoning process. This setting aligns with prior findings that LLMs are generally unable to refine their own solutions without external feedback (Huang et al., 2023; Valmeekam et al., 2023). For test-time scaling, we reproduced the sequence test-time scaling (Muennighoff et al., 2025) and observed clear performance gains, as shown in Figure 5.
>
> Huang, J., Chen, X., Mishra, S., Zheng, H. S., Yu, A. W., Song, X., & Zhou, D. (2023). Large language models cannot self-correct reasoning yet. arXiv preprint arXiv:2310.01798.
>
> Valmeekam, K., Marquez, M., & Kambhampati, S. (2023). Can large language models really improve by self-critiquing their own plans? arXiv preprint arXiv:2310.08118.
>
> Niklas Muennighoff, Zitong Yang, Weijia Shi, Xiang Lisa Li, Li Fei-Fei, Hannaneh Hajishirzi, Luke Zettlemoyer, Percy Liang, Emmanuel Candes, and Tatsunori Hashimoto. s1: Simple test-time scaling. arXiv preprint arXiv:2501.19393, 2025.

---

### Official Review · Reviewer_Yx6b · 2025-10-31

**Soundness:** 2
**Presentation:** 3
**Contribution:** 3
**Rating:** 6
**Confidence:** 4

**Summary:**

This paper introduced critique reinforcement learning (CRL). It is a training paradigm designed to improve models' reasoning and reflective abilities for code generation tasks. It  explicitly trains model to  critique and judge the correctness of generated solutions.
So in addition to traditional outcome-based signal, they introduced a new signal regarding the correctness on question-solution paris. The experiments on several code generation tasks show improvements after leveraging critique reinforcment learning.

**Strengths:**

1. This paper introduced a new perspective which is to leverage critique signals to improvement the performance during reinforcement learning.
2. Tis work uses test case execution results to automatically generate judgement labels, enabling scalable data creation without human annotation.

**Weaknesses:**

1. The improvement of 8B model is limited. For example, on LiveCodeBench, it improves 1.2% and on Aider-Polyglot, it improves 1.1%. Setting aside noise and variance, I observe that as the model size increases, the performance gain diminishes. This raises concerns about whether the proposed method can effectively scale up.

2. For CRL process, where are the groundtruth judgement label c* from? From Figure 3 and page 4, it looks to me these judgement labels are from the text case execution and its pass rate. However, during normal RL training, we  have already had such signals from pass rate of each rollout solution. The proposed method just make it happen earlier and separately. I wonder how much gains could obtain from this process and what's the real source of improvement.

3. In the current implementation, how do you aggregate two rewards: reward from RL and reward from CRL? Since multi-objective RL is not very robust, Is there any instability or robustness issue with different aggregation methods?

**Questions:**

N/A

---

> ### Author Response · Authors · 2025-11-14
>
> We thank the reviewer for the helpful comments and constructive evaluation. We are glad that the contributions and presentation were well received, and we address the concerns point-by-point below.
>
> R1: Scalability of CRL Across Model Sizes
> We acknowledge that the improvement on Aider-Polyglot for the 8B model is modest; however, it is worth noting that CRL yields a +2.7% gain on BigCodeBench-I Hard. Notably, in our CRL data, the candidate solutions were generated by Qwen3-Coder-30B-A3B-Instruct (non-thinking), which is weaker than Qwen3-8B (thinking). As shown in the table below, Qwen3-Coder-30B-A3B-Instruct even underperforms Qwen3-4B (thinking) on our training dataset and LiveCodeBench v5, confirming that the candidate generator is relatively weak. Such weak candidate solutions provide only limited learning signals; nevertheless, the 8B model still achieves consistent gains, indicating the effectiveness and robustness of the CRL approach.
>
> | Model                           | Pass@1 on Training Dataset | LiveCodeBench v5 |
> |---------------------------------|-----------------------------|------------------|
> | Qwen3-Coder-30B-A3B-Instruct    | 32.20%                      | 40.7             |
> | Qwen3-4B (thinking)             | 43.72%                      | 54.2             |
>
> R2: On the Source and Role of Critique Labels
> The ground-truth critique labels c∗ are indeed derived from test case execution results and corresponding pass rates, as noted by the reviewer. However, unlike standard RL, where the pass rate directly serves as a reward for the final solution, CRL explicitly trains the model to judge whether a given candidate solution is correct. This additional reward enables the model to internalize evaluation and reasoning skills, allowing it to form a critique-based understanding of correctness rather than relying solely on outcome-based feedback. In this way, CRL complements traditional RL by encouraging reflective reasoning beyond simple reward optimization.
>
> R3: On Reward Aggregation and Stability
> We experimented with different reward scaling coefficients to balance the RL and CRL objectives. Specifically, during the 16k-token training stage, we applied a scaling factor of 0.8 to the CRL reward to match the magnitude of the two signals. At the 32k stage, both rewards were used with their original scales. This dynamic adjustment maintained stable training and prevented dominance from either objective, and we did not observe instability or robustness issues under this setup.

---

### Official Review · Reviewer_badP · 2025-11-01

**Soundness:** 3
**Presentation:** 3
**Contribution:** 2
**Rating:** 4
**Confidence:** 4

**Summary:**

This paper presents Critique Reinforcement Learning (CRL), a training paradigm designed to teach large language models to critique solutions rather than merely generate them. Given a (question, solution) pair, the model produces a critique containing a final binary judgment  c∈{True,False}, which is rewarded based on whether it matches the ground-truth correctness label . The authors further propose CRITIQUE-CODER, trained with 80% standard RL data and 20% CRL data. Empirical evaluations on EvalPlus, BigCodeBench, LiveCodeBench v5, and BBEH benchmarks show that CRITIQUE-CODER consistently outperforms RL-only baselines, including DeepCoder-14B and GPT-o1, especially in code reasoning tasks.
While the paper is well-executed and clearly written, its claim of being “verifier-free” and “self-improving” is only partially justified, since CRL fundamentally depends on verifiable signals derived from executable test cases and strong model–generated solutions.

**Strengths:**

1. Addresses a timely question: how to endow models with critique and reflection abilities.

2. Clear algorithmic presentation (Algorithm 1) and reproducible training setup.

3. Comprehensive experiments across multiple code benchmarks.

4. Demonstrates that mixing 20% CRL data improves robustness and mitigates policy collapse.

5. Writing quality, figures, and clarity of exposition are strong.

**Weaknesses:**

1. Dependence on verifiable signals and strong model–generated solutions.
The ground-truth labels
are obtained through executable test cases with a pass-rate threshold (80%), and the candidate solutions are generated by a stronger model (Qwen3-30B). Although no external verifier module is used, this still constitutes a verifiable supervision pipeline, limiting applicability to domains where executable feedback is available.

2. Limited differentiation from prior critique-based methods.
CRL is conceptually close to CFT and CGD, differing mainly in the reinforcement reward design. The absence of direct experimental comparisons leaves its unique contribution ambiguous.

3. Sparse reward and lack of training dynamics analysis.
The reward is binary (0/1), yet the paper provides no discussion of variance, convergence stability, or reward sensitivity to the pass-rate threshold.

4. “General reasoning” claims rely on narrow evidence.
Improvements on the BBEH dataset are encouraging but cover only one reasoning benchmark. There is no evidence of gains in non-verifiable or natural-language reasoning domains.

5. Absence of self-critique capability (acknowledged by authors).
Section 3.7 explicitly reports that enabling the model to critique its own outputs does not improve performance, suggesting the learned skill is evaluative (“other-critique”) rather than reflective (“self-critique”).

**Questions:**

1. How would CRL perform if the candidate solutions were generated by a weaker or equal-capacity model?

2. Have you attempted CRL in non-code domains lacking executable verification?

3. Could you provide training curves or reward variance to assess stability under binary rewards?

4. What differences exist between your approach and CFT/CGD when controlling for compute and data volume?

5. Why do self-critique variants fail—does this relate to policy–critic decoupling or distributional mismatch?

---

> ### Author Response · Authors · 2025-11-14
>
> We thank the reviewer for the thoughtful and constructive feedback and appreciate the recognition of our work’s clarity, experimental design, and overall contribution.
>
> R1: Effect of Critiques from Weaker Models
> The model used to generate candidate solutions, Qwen3-Coder-30B-A3B-Instruct (non-thinking), is not stronger than Qwen3-4B (thinking), as it lacks reasoning capability. This is evidenced by its lower performance on both our training dataset and LiveCodeBench, where Qwen3-4B (thinking) achieves higher scores. The 30B model’s weaker performance indicates that the observed improvement cannot stem from knowledge distillation or teacher strength. Even when the critiques are derived from these weaker solutions, our CRL approach still leads to consistent gains, demonstrating its robustness and effectiveness.
>
> **Model Strength Comparison**
> | Model                          | Pass@1 (Train) | LiveCodeBench v5 |
> |--------------------------------|----------------|------------------|
> | Qwen3-Coder-30B-A3B-Instruct   | 32.20%         | 40.7             |
> | Qwen3-4B (thinking)            | 43.72%         | 54.2             |
>
> R2: CRL in Non-Executable Settings
> We have not applied CRL to non-code domains that lack executable verification. In such settings, obtaining ground-truth correctness for candidate solutions is inherently difficult, as there is no reliable way to determine whether a given solution is correct. Without executable feedback, it becomes challenging to evaluate the quality of critiques or assign meaningful rewards based on their accuracy.
>
> R3: Training Stability under Binary Rewards
> We report the mean and variance of the reward over the first 40 training steps for the Qwen3-4B model to illustrate early training stability. The results show that the reward distributions of CRL and standard RL are largely similar. To maintain a balanced reward scale between the two data types during 16k-length training, we applied a scaling coefficient of 0.8 to the CRL rewards.
>
> **Reward Mean and Variance (R3)**
> | Method | Mean   | Variance |
> |--------|--------|----------|
> | CRL    | 0.4820 | 0.0013   |
> | RL     | 0.4749 | 0.0015   |
>
> R4: Comparison with CFT/CGD
> Both CFT and CGD train models to imitate critique traces under a supervised fine-tuning (SFT) setup, rather than providing rewards based on the correctness of critique judgments. In contrast, our CRL approach explicitly learns from reward feedback, enabling the model to refine its critique ability through reinforcement rather than imitation.
> To verify this distinction, we conducted CFT experiments by using Qwen2.5-32B-Instruct to generate candidate solutions and GPT-4.1-mini to produce critiques, training a Qwen2.5-Coder-7B model on this data. The resulting performance was lower than the baseline, indicating that merely mimicking critique traces (as in CFT) does not yield improvement. This demonstrates that CRL and CFT fundamentally differ in their learning mechanisms—the former emphasizes reward-driven judgment accuracy, while the latter focuses on behavioral imitation.
>
> **CFT vs Baseline**
> | Model                | LiveCodeBench v5 |
> |----------------------|------------------|
> | Qwen2.5-Coder-7B     | 24.6             |
> | Qwen2.5-Coder-7B-CFT | 21.2             |
>
> R5: The failure of self-critique
> The failure of self-critique variants is not a new phenomenon; prior work has shown that, without external feedback, models struggle to correct their own erroneous solutions (Huang et al., 2023; Valmeekam et al., 2023). Our findings are consistent with these observations. The improvements brought by CRL do not stem from a model critiquing its own outputs, but rather from learning to internalize critique-based reasoning through reward-driven supervision. This leads to genuine enhancement of reasoning ability, rather than self-correction within the same policy.
>
> Huang, J., Chen, X., Mishra, S., Zheng, H. S., Yu, A. W., Song, X., & Zhou, D. (2023). Large language models cannot self-correct reasoning yet. arXiv preprint arXiv:2310.01798.
>
> Valmeekam, K., Marquez, M., & Kambhampati, S. (2023). Can large language models really improve by self-critiquing their own plans? arXiv preprint arXiv:2310.08118.

---

### Official Review · Reviewer_jfJs · 2025-11-03

**Soundness:** 2
**Presentation:** 2
**Contribution:** 2
**Rating:** 4
**Confidence:** 4

**Summary:**

This paper proposes training code generation models on a mixture of two tasks: standard code generation and critique evaluation. The critique task involves judging whether a given solution correctly solves a problem, receiving binary rewards based on judgment accuracy. The authors train Critique-Coder using GRPO on 80% code generation data and 20% critique data, fine-tuning Qwen3-4B and Qwen3-8B models. Results show improvements across coding benchmarks, with Critique-Coder-8B achieving 60.8% on LiveCodeBench (v5). The approach also improves performance on BBEH logical reasoning tasks, suggesting the critique task develops transferable reasoning skills beyond coding.

**Strengths:**

- Comprehensive evaluation across diverse benchmarks (LiveCodeBench, EvalPlus, Aider-Polyglot, BigCodeBench) demonstrates consistent improvements over single-task training baselines.
- Evidence of transfer to general reasoning (BBEH benchmark) suggests the critique task develops broadly applicable skills rather than narrow task-specific capabilities.
- Ablation study provides practical guidance, showing 20% critique data mixing is optimal. Too much critique data (50-100%) degrades performance due to train-test mismatch.

**Weaknesses:**

- Critique training data is generated using a different model (Qwen3-Coder-30B-A3B-Instruct) rather than the model's own solutions. This creates a fundamental distribution mismatch, the model learns to critique a 30B model's solution distribution, not its own, which likely explains the complete failure of test-time self-critique in Section 3.7.
- The paper is missing the obvious experiment which would be on-the-fly critique generation where models critique their own on-policy solutions during training. The paper currently takes a shortcut using pre-generated data, that too from a different model.
- Confusing terminology—"CRL" and "RL" both refer to GRPO training on different tasks (critique vs. generation). This is fundamentally a  multi-task learning setup rather than a novel training algorithm.
- Due to lack of proper ablations, it's hard to discern the reasons for the performance boost. The observed improvements could stem from multiple confounds: simple multi-task learning effects, knowledge distillation from the 30B model, or just increased training compute. The paper doesn't isolate the actual source of gains.

**Questions:**

- Why generate critique data from a different 30B model instead of using the training model's own solutions? Did you consider on-the-fly critique generation during training?
- Have you investigated whether the test-time critique failure is specifically due to the distribution mismatch between training critiques (from 30B model) and test-time critiques (of own solutions)?
- Can you provide ablations isolating the source of improvements: is it multi-task learning, distillation from the larger model, or genuine critique capability?

---

> ### Author Response · Authors · 2025-11-14
>
> We thank the reviewer for the helpful comments and for highlighting both strengths and areas for clarification. Below, we respond to the concerns about critique data sources, test-time failure analysis, and ablations.
>
> R1. On the use of critique data from a different model (Qwen3-Coder-30B-A3B-Instruct)
> We appreciate the reviewer’s concern. Notably, the 30B model used to generate critiques Qwen3-Coder-30B-A3B-Instruct (non-thinking) actually performs worse than our base models, Qwen3-4B (thinking) and Qwen3-8B (thinking), as the Coder model does not support the “thinking” mode. Without extended reasoning traces, the 30B model achieves lower pass@1 accuracy on our training datasets and lower LiveCodeBench scores than Qwen3-4B (thinking). Thus, the 30B model is not a teacher in our setup, and there is no distillation involved. It merely serves as a source of diverse judgment signals.
>
> | Model                           | Pass@1 on Training Dataset | LiveCodeBench v5 |
> |---------------------------------|-----------------------------|------------------|
> | Qwen3-Coder-30B-A3B-Instruct    | 32.20%                      | 40.7             |
> | Qwen3-4B (thinking)             | 43.72%                      | 54.2             |
>
> We agree that on-the-fly critique generation during training is a promising direction. Although we did not implement full on-policy critique generation in this work, we conducted a related experiment where Qwen3-4B was trained using its own generated candidate solutions under the same configuration (16K output length, 20% critique data, 40 training steps). As shown in the table below, this self-generated variant (Qwen3-4B-CRL-40steps) improved LiveCodeBench performance from 54.2 → 56.6, outperforming both the baseline and the RL-only variant (54.8). This indicates that the critique task remains effective even without relying on a different model’s solution distribution, and it suggests that full on-the-fly critique generation would likely yield additional gains in future work.
>
> | Model                 | LiveCodeBench v5 |
> |-----------------------|------------------|
> | Qwen3-4B (thinking)   | 54.2             |
> | Qwen3-4B-CRL-40steps  | 56.6             |
> | Qwen3-4B-RL-40steps   | 54.8             |
>
> R2. On whether the test-time critique failure is caused by distribution mismatch
> First, the 30B model used for candidate solution generation is not stronger than Qwen3-4B (thinking). Second, recent studies have shown that large language models cannot critique or refine their own outputs without external information (Huang et al., 2023; Valmeekam et al., 2023). Therefore, the failure of self-critique at test time is expected and not indicative of a training–test mismatch. We argue that the benefit of critique training lies in internalizing reasoning patterns rather than directly critiquing one’s own generations.
>
> Huang, J., Chen, X., Mishra, S., Zheng, H. S., Yu, A. W., Song, X., & Zhou, D. (2023). Large language models cannot self-correct reasoning yet. arXiv preprint arXiv:2310.01798.
>
> Valmeekam, K., Marquez, M., & Kambhampati, S. (2023). Can large language models really improve by self-critiquing their own plans? arXiv preprint arXiv:2310.08118.
>
> R3 On isolating the source of improvements
> First, we can rule out distillation from a larger model as a source of improvement, since the 30B model does not outperform Qwen3-4B-(thinking) on either our training dataset or LiveCodeBench. To further distinguish between multi-task effects from genuine critique capability, note that the candidate solutions in CRL’s training data are actually generated by weaker models, making it unlikely for the model to learn effectively from them; thus, the multi-task explanation appears implausible. Moreover, we evaluate our models on the BIG-Bench Extra Hard (BBEH) logical reasoning benchmark, which is unrelated to coding. Models trained with critique supervision consistently outperform pure RL baselines on these reasoning tasks, suggesting that the improvements arise from enhanced critique-based reasoning ability rather than from generic multi-task regularization.
>
> R4. On whether CRL is merely multi-task learning
> Since the model used to generate candidate solutions is not stronger than our base models, the training process cannot be viewed as distillation or knowledge transfer.
> The model does not learn additional problem-solving content from the provided solutions; in fact, it even benefits from analyzing and correcting weaker solutions.
> This indicates that the improvement arises from learning how to evaluate and reason about solutions, rather than simply optimizing for multiple objectives.
> Therefore, the proposed CRL framework is fundamentally not a multi-task learning setup, but a reward-coupled reasoning process, where critique supervision shapes the generation policy toward verification-oriented behavior.

---

### Comment · Area_Chair_zgHt · 2025-11-26

Dear Reviewers,

Would you please check authors' rebuttal and see if they have addressed your comments?

Best

AC

---

### Author Response · Authors · 2025-12-02

**Major Point 1 — CRL’s performance gains are not due to distillation**

Reviewer jfJs and badP proposed two alternative explanations for the improvements of Critique Reinforcement Learning (CRL):
(1) implicit distillation from the model used to generate critique data, and
(2) multi-task training effects.
Both interpretations are based on incorrect assumptions.

First, the distillation argument incorrectly assumes that Qwen3-Coder-30B-A3B-Instruct is stronger than the Qwen3-4B/8B (thinking) models. In fact, it is substantially weaker due to the absence of the thinking-mode reasoning process. As shown in Table 1, the 30B model underperforms Qwen3-4B (thinking) by 11.52 Pass@1 points on our training set and 13.5 points on LiveCodeBench v5. These numerical differences make it clear that the 30B model cannot serve as a stronger "teacher", thus ruling out implicit distillation as a plausible explanation. Importantly, CRL still improves even when the critiques are generated by this weaker 30B model, further confirming that its gains arise from genuine critique-based policy improvement rather than inheriting superior behaviors from an external model.

Table 1. Comparison between 30B and 4B models

| Model                           | Pass@1 on Training Dataset | LiveCodeBench v5 |
|---------------------------------|-----------------------------|------------------|
| Qwen3-Coder-30B-A3B-Instruct    | 32.20%                      | 40.7             |
| Qwen3-4B (thinking)             | 43.72%                      | 54.2             |

Second, CRL is not a multi-task training setup. As stated in Section 2.2 Dataset Construction, CRL simply replaces 20% of the original RL data with critique-evaluation data while holding the total training volume constant. There is no secondary task or auxiliary prediction objective.

Most importantly, CRL improves performance even when it critiques its own solutions, i.e., without any external model involved. As shown in Table 2, Qwen3-4B-CRL achieves 56.6, outperforming both the RL baseline (54.8) and the original (54.2) model:

Table 2. CRL improves the model using its own solutions.

| Model                 | LiveCodeBench v5 |
|-----------------------|------------------|
| Qwen3-4B (thinking)   | 54.2             |
| Qwen3-4B-CRL-40steps  | 56.6             |
| Qwen3-4B-RL-40steps   | 54.8             |

This improvement emerges without any stronger model, and under identical training volume, confirming that CRL introduces a genuine critique-based policy improvement mechanism rather than inheriting behavior from a more capable teacher.

&nbsp;

**Major Point 2 — Self-critique failure is not a novel finding**

Reviewer jfJs, badP, and qbh7 all raised concerns regarding the experiment presented in Section 3.7: Limitations in Self-Critique, noting that in this experiment, the model only sees the final answer rather than the underlying reasoning trajectory. We would like to clarify that this limitation is well-known in prior literature and not a new or surprising phenomenon specific to our method.

As discussed in our paper Section 4.1: Critique Learning of the Related Works, several influential works have already demonstrated that models cannot reliably refine or correct their own outputs without access to additional useful external signals. These studies consistently show that, when a model is given only its final solution, without intermediate reasoning steps, its self-evaluation is fundamentally unreliable. For convenience, we cite again the two key papers referenced in our Related Work section here.

Huang, J., Chen, X., Mishra, S., Zheng, H. S., Yu, A. W., Song, X., & Zhou, D. (2023). Large language models cannot self-correct reasoning yet. arXiv preprint arXiv:2310.01798.

Valmeekam, K., Marquez, M., & Kambhampati, S. (2023). Can large language models really improve by self-critiquing their own plans? arXiv preprint arXiv:2310.08118.

Given this context, the observation that our model fails to self-critique its own outputs in this constrained setting is fully aligned with prior findings. Importantly, our results indicate that the benefits of CRL do not come from post-hoc self-critique on final outputs. Instead, CRL improves performance because the model internalizes critique-based judgment during training, enabling it to perform more thorough reasoning and refinement before producing the final answer. Consequently, once deprived of the reasoning chain, the model naturally cannot further critique or refine the final result, consistent with established limitations in the field.

---

### Meta-Review · Area_Chair_gPAF · 2026-01-13

**Summary:**

The paper proposes Critique Reinforcement Learning (CRL), a training paradigm that augments standard RL for code generation. In CRL, the model is trained not only to generate solutions but also to critique (judge correctness of) (question, solution) pairs. The authors introduce Critique-Coder, trained on a hybrid mixture (80% standard RL, 20% CRL) using the GRPO algorithm. Experiments on benchmarks like LiveCodeBench, EvalPlus, and BigCodeBench show that Critique-Coder outperforms RL-only baselines and demonstrates transferable reasoning abilities on the BBEH logical reasoning dataset.

**Reviewer Concerns:**

Addressed Concerns:

The primary concern regarding "implicit distillation" (Reviewers jfJs, badP) was conclusively refuted; the authors demonstrated that the 30B "teacher" model was actually weaker than the 4B base model, ruling out knowledge transfer as the source of gains. Additionally, the concern about data distribution mismatch was addressed by new experiments showing that training on self-generated ("on-policy") solutions yields similar improvements to the proposed method.

Outstanding Concerns:

The model's inability to effectively self-critique at test time remains a limitation, though the authors correctly note this is a known issue in current LLM literature. Additionally, the concern regarding scalability (Reviewer Yx6b)—specifically the diminishing returns observed on the 8B model compared to the 4B model—remains an open question for future scaling.

**Reviewer Scores:**

Current Scores: 8, 6, 4, 4

Reviewers qbh7 (8) and Yx6b (6) would likely maintain their positive scores.

Reviewers badP (4) and jfJs (4) would likely have raised their scores to 6 had they fully engaged with the rebuttal. Their primary technical objection was the "distillation" hypothesis. The authors conclusively refuted this in the rebuttal by demonstrating that the 30B "teacher" model is significantly weaker than the 4B base model on the target benchmarks. Since the teacher is weaker, distillation cannot explain the gains. Furthermore, the authors provided the specific ablation requested by Reviewer jfJs (training on self-generated solutions), showing that the method remains effective. Since the foundational reasons for the low scores were factually addressed, an increase in score is the logical outcome.

---

### Decision · Program_Chairs · 2026-01-26

Accept (Poster)